# Interrupted Nef and Meyer Reactions: A Growing Point for Diversity-Oriented Synthesis Based on Nitro Compounds

**DOI:** 10.3390/molecules28020686

**Published:** 2023-01-10

**Authors:** Alexey Yu. Sukhorukov

**Affiliations:** N. D. Zelinsky Institute of Organic Chemistry, Russian Academy of Sciences, Leninsky Prospect, 47, Moscow 119991, Russia; sukhorukov@ioc.ac.ru

**Keywords:** interrupted reactions, Nef reaction, Meyer reaction, nitro compounds, diversity-oriented synthesis, nucleophilic addition, iminium cations, nitrile oxides

## Abstract

The Nef reaction (nitro to carbonyl group conversion) and related Meyer reaction are among the key transformations of aliphatic nitro compounds. The interrupted versions of these reactions in which the normal pathway is redirected to a different end product by an external nucleophile are much less common, albeit these processes substantially increase the synthetic potential of nitro compounds. In this review, examples of interrupted Nef and Meyer reactions are summarized, and the prospects of this methodology in diversity-oriented organic synthesis are analyzed. The bibliography contains 90 references.

## 1. Introduction

Nitro to carbonyl group conversion is one of the key transformations in organic synthesis known since the second half of the 19th century. The action of strong acids on primary nitroalkanes leading to carboxylic acids was first documented by Meyer and Wurster in 1873 [1]. The conversion of nitroalkane salts into aldehydes and ketones through acid hydrolysis (known as the Nef reaction) was discovered independently by M. I. Konovalov in 1893 [2] and J. Nef in 1894 [3]. In the following decades, these reactions received much development both in terms of methodology and application in the synthesis of complex natural molecules [4,5,6,7,8,9,10,11,12,13,14,15,16,17,18,19,20,21,22,23,24,25,26,27]. Of the most important examples where the nitro to carbonyl group conversion is used, the method of chain elongation in carbohydrates (the Fischer–Sowden method) can be mentioned [6,17].

The classical Nef reaction involves the deprotonation of a nitro compound to give a nitronate anion followed by its treatment with an aqueous solution of a protic acid [8,19]. Although reductive [13] and oxidative [15,16] versions have also been developed, the hydrolytic Nef reaction has not lost its significance and is still frequently used in modern organic synthesis [20].

The generally accepted unified mechanism of the Nef and Meyer reactions is believed to involve the *N*,*N*-bis(hydroxy)iminium cation **I-1** as a common intermediate, which is generated by the protonation of a nitronic acid (or double protonation of a nitronate anion, Figure 1) [9,10,11,12,19,28]. In the Nef reaction, the nucleophilic addition of a water molecule to cation **I-1** occurs followed by the elimination of HNO to give a ketone or aldehyde. In the Meyer reaction, which requires stronger acid conditions [8,12], cation **I-1** eliminates water to give a nitrosocarbenium cation **I-2** or tautomeric protonated hydroxynitrilium cation **I-3**. Upon addition of water, these cations are converted to a hydroxamic acid, which is then hydrolyzed to a carboxylic acid. Hydration of nitrosocarbenium cation **I-2** can also result in a Nef product (aldehyde) if the tautomerization of the transient *gem*-hydroxynitroso compound is slow.

In principle, cationic species **I-1**, **I-2,** and **I-3** can be intercepted by nucleophiles other than water (such as halide anions, alcohols, carboxylic acids, thiols, amines, electron-rich arenes, etc.) resulting in the interruption of the normal Nef/Meyer reaction pathway. This would lead to either α-substituted nitroso compounds or oximes as the end products (Figure 1). In this way, a set of new transformations forming carbon–carbon and carbon–heteroatom bonds could be implemented, in which the nitro group is converted to other functions with activation by protic acids. Note that the traditional polarity of the nitro moiety as an α-C-nucleophile is reversed in these reactions [29,30,31].

The recently introduced concept of interrupted reactions was not applied to Nef and Meyer reactions previously [32]. Moreover, the implementation of the interrupted versions of these reactions is complicated in practice since the majority of common nucleophiles are incompatible with protic acids. Nevertheless, a literature survey revealed a number of reactions that could be classified as interrupted Nef/Meyer reactions. This mini-review summarizes these scattered examples and analyzes the prospects of this methodology for the development of diversity-oriented synthesis based on readily available nitroalkanes as starting materials. Note that only processes promoted by protic acids and hydrogen bonding are considered here; the electrophilic activation of nitroalkanes is covered in recently published reviews [30,33,34,35]. Moreover, the acid-mediated transformations of nitroalkenes and nitroarenes are outside of the scope of this mini-review [36].

The material in this review is systemized according to the nature of the nucleophilic partner that induces the termination of the normal Nef reaction pathway: (1) halide anions; (2) *O*- and *S*-nucleophiles; (3) *N*-nucleohiles; and (4) *C*-nucleophiles. Specific focus is given to intramolecular reactions leading to valuable heterocyclic frameworks and the total synthesis of natural products using interrupted Nef/Meyer reactions.

## 2. Interruption of the Nef and Meyer Reactions by Halide Anions

Soon after the discovery of the Nef reaction, Steinkopf and Jürgens reported that the reaction of nitroethane sodium salt **1a** with gaseous HCl produced a blue-colored compound, for which the structure of 1-chloro-1-nitrosoethane **2a** was suggested (Figure 2a) [37]. The latter underwent subsequent conversion into colorless acethydroxamyl chloride **3a**. This report seems to be the first observation of an interrupted Nef process.

Later, Nametkin [5], Nenitzescu [40], Arndt [41], Kornblum [10], and other researchers [8,42] observed the formation of hydroxamic acid chlorides from primary nitroalkanes in the Nef reaction using hydrochloric acid, albeit often in low yields. The formation of intermediate labile blue-colored nitroso species was observed in many cases. For example, Gil and MacLeod [38] reported that the reaction of the sodium salt of (3-nitropropyl)benzene **1b** with HCl in ether at −60 °C produced a blue-colored chloronitroso compound **2b**, which dimerized upon crystallization (Figure 2b).

A number of examples of successful preparation of hydroxamyl chlorides from primary aliphatic nitro compounds were reported [39,43,44]. In particular, allylhydroximoyl chloride **3c** and 4-methylthiobutylhydroximoyl chloride **3d** were generated in the reactions of sodium salts of the corresponding nitro compounds **1c**,**d** with HCl in dioxane at −78 °C [39]. The resulting hydroximoyl chlorides were not isolated and were used in the reaction with ^13^C-labelled thioglucose derivative **4** to afford glucosinolates **5c**,**d** (Figure 2c).

Yao et al. [45,46] developed a convenient one-pot synthesis of hydroxamyl halides **3** from nitrostyrenes **6** by the addition of organometallic reagents followed by treatment of the resulting nitronate anions with a concentrated HHal solution (Figure 3). Remarkably, hydroxamyl iodides (Hal = I) could be prepared by this method. Unlike chlorides **3**, the yields of the corresponding bromides and especially iodides substantially decreased with an increase in the steric hindrance of the nitronate. The formation of stable nitrile oxides **7** as major products was observed in the case of very bulky nitronates, suggesting that hydroxynitrilium cations (Meyer reaction intermediates **I-3**, Figure 1) are common intermediates in the formation of products **3** and **7**. However, the appearance of transient blue or green colors in these reactions is indicative of halonitroso compounds that are generated by the addition of a halide anion to cations **I-1** and **I-2**. Thus, in principle, both interrupted Nef and Meyer mechanisms may operate here.

β-Nitroketones **8** react with HHal in acetic acid to produce 3-haloisoxazoles **10** as reported by Fusco and Rossi [47] and Carr et al. [48] (Figure 4). These products are formed through the conversion of the nitro compound into hydroxamic acid halide **9** followed by cyclization. 3-Chloroisoxazoles were prepared in moderate yields, while the formation of the corresponding bromides was less efficient. Additionally, substrates **9** with donor aryl groups gave products **10** in lower yields. The initial β-nitroketones are prepared by Friedel–Crafts acylation of arenes with 3-chloropropanoyl chloride followed by the Kornblum reaction.

Recently, our group reported an interrupted Nef reaction with 5- and 6-membered cyclic nitronic esters **11** (Figure 5) [49]. The treatment of the latter with HCl in dioxane resulted in the ring opening and formation of chloronitroso compounds **12** bearing a distant hydroxyl group. Remarkably, the process was usually stereoselective. Moreover, in some cases, the stereochemical outcome depended on the temperature, thus allowing a stereodivergent synthesis of products **12** to be performed (see products **12a** and **12b** in Figure 5). DFT computations confirmed that the most likely mechanism involves the protonation of the exocyclic oxygen atom to form the *N*,*N*-bis(oxy)iminium cation **I-4** (a common Nef intermediate), which is converted to the product by nucleophilic addition of a chloride anion and ring opening in the resulting nitrosoacetal **I-5**.

## 3. Interruption of the Nef and Meyer Reactions by *O*- and *S*-Nucleophiles

Numerous examples of the Nef and Meyer reactions in which a protonated nitro moiety is attacked by a hydroxyl, carboxylic group, or *O*-enolate are documented in the literature. All of these transformations correspond to intramolecular processes that occur faster than the parent reaction with water as the nucleophile.

Meinwald et al. observed that nitrolactone **13** under standard Nef conditions (treatment with sodium methylate followed by methanolic sulfuric acid) afforded cyclic acetal **14** instead of the expected aldehyde (Figure 6) [50]. The obtained compound most likely results from the initial methanolysis of the lactone with sodium methoxide to give a nitronate hydroxy ester followed by an intramolecular cyclization involving the hydroxyl group and protonated nitronic acid moiety. Acetal **14** was then utilized as an intermediate in the total synthesis of *rac*-pederamide.

The Nef-type reaction of nitrodiols **15** is a well-established strategy for spiro- and bicycloketals that are widely present in natural sources (Figure 7) [20]. A common procedure involves the conversion of a nitrodiol into a corresponding nitronate salt followed by treatment with aqueous hydrochloric or sulfuric acid. Since ketodiols are not isolated in this process, it is logical to assume that the first cyclization takes place before the hydrolysis of the *N*,*N*-bis(hydroxyl)iminium cation **I-6**, e.g., an interrupted Nef reaction occurs. The subsequent elimination of HNO provokes the second cyclization, leading to the assembly of the ketal moiety. This methodology was successfully utilized by Ballini et al. and Guarna et al. to build 5,5- [51,52,53], 5,6- [54,55], 6,6- [56,57], and 6,7- [55,56] membered spiroketals as exemplified in Figure 7a–c. The total synthesis of several natural compounds (pheromones and components of odors of insects) was accomplished via the cyclization of nitrodiols. The accessibility of chiral δ-nitroalcohols via asymmetric reduction of corresponding nitroketones allows the synthesis of enantiopure spiroketals [52,53].

Recently, Hong et al. developed an organocatalytic enantioselective strategy to construct a benzene-fused bicycloketal moiety, which is a common motif in aflatoxins (Figure 8) [58,59]. Here, nitroalkenes **16** were involved in an asymmetric Michael addition with aldehydes with 20 mol % of Jørgensen−Hayashi catalyst to give the corresponding nitroaldehydes. The latter were reduced to nitroalcohols with NaBH_4_ followed by a standard hydrolytic Nef protocol to give the desired tricyclic acetals **17**. The aforementioned sequence was performed in a one-pot manner leading to good yields of products **17** and high levels of diastereo- and enantiocontrol. The developed strategy was successfully utilized in the synthesis of natural coumarin (−)-microminutinin [58].

The carboxylic group can play the role of an intramolecular *O*-nucleophile in the Nef reaction (Figure 9). Interestingly, an external acid is often not needed in these reactions, which proceed under neutral or basic conditions via the cyclic intermediates of type **I-7**. In fact, Wilson and Lewis observed that 4-nitrovaleric acid **18** underwent a spontaneous Nef reaction, while 3-nitropropionic and 6-nitrohexanoic acids were stable to hydrolysis (Figure 9a) [60]. Such a dramatic acceleration effect was rationalized by the formation of 5-membered cyclic intermediate **I-8** from the *aci*-form of 4-nitrovaleric acid. The process resulted in a normal Nef product (levulinic acid **19**), while no products resulting from interrupted Nef intermediate **I-8** were detected. A similar acceleration of the Nef reaction was observed in the hydrolysis of substituted 4-nitrovaleric and 4-nitrohexanoic acids [61].

In the case of primary nitroalkanes, the cyclic intermediate **I-7** is not hydrolyzed due to faster dehydration leading to interrupted Meyer reaction products. For example, Keumi et al. reported that benzoic acids **20** bearing a nitromethyl group in the *ortho*-position underwent cyclization to *N*-hydroxyisophthalimides **21** after treatment with a cold aqueous Na_2_CO_3_ solution (Figure 9b) [62]. The suggested mechanism for the cyclization involved the generation of a nitronic acid that underwent cyclization with the adjacent carboxylic group and the elimination of a water molecule. Upon heating, the products **21** rearranged to *N*-hydroxyphthalimides **22**.

A similar cyclization of phosphonate-substituted 4-nitrobutanoic acid **23** was reported by Krawczyk et al. (Figure 9c) [61]. The formation of *N*-hydroxyimide **24** was accomplished by refluxing in water for 2h. 3-Aryl-substituted acids of type **23** exhibited the same reactivity [63].

Peters et al. reported a related interrupted Meyer-type reaction, in which the process was redirected to another end product by intramolecular cyclization with the benzamide moiety in nitroamide **25** upon treatment with a base (NaOAc) in the presence of AcOH and Ac_2_O (Figure 10) [64]. As a result, 1,3-oxazinan-6-one oxime ester **26** was formed in a moderate yield. Since the reaction was performed in the presence of acetic anhydride, it is difficult to conclude whether *O*-protonated or *O*-acylated iminium species served as intermediates. Additionally, the intermediacy of a nitrile oxide in this reaction cannot be ruled out, since these species are known to form by treatment of primary nitroalkanes with acid anhydrides. However, trapping experiments with methylacrylate gave no 1,3-dipolar cycloaddition products. A similar cyclization was also reported by Kazmaier et al. [65].

Enolates were shown to act as *O*-nucleophiles in the intramolecular cyclization of Nef and Meyer intermediates. Like with carboxylate nucleophiles, an acid medium is not needed to induce Nef or Meyer reactions here. This shows a strong neighboring group effect of the enol moiety in these substrates.

Thus, nitrodiketones **27** generated by the reaction of 1,3-cyclohexanedione or 1,3-cycloheptanedione with nitrostyrenes undergo cyclization to give bicyclic furan-2(3*H*)-one oximes **28** (Figure 11a) [66,67,68]. Under the same conditions, the reaction of α-methyl nitrostyrene with dimedone afforded an annulated furan derivative **29**. The formation of this product can be interpreted as the result of an interrupted Nef reaction followed by HNO elimination (Figure 11b) [69].

Ibrahim et al. reported a somewhat related cyclization of 3-nitroacetylquinolinone **30** to the furo [3,2-c]quinolinone derivative **31** (Figure 11c) [70]. Drastic conditions (reflux in DMF) were required for this process.

Recently, Shen et al. developed an enantioselective synthesis of tetrahydrobenzofuranone oximes **33** by a squaramide-catalyzed Michael addition/cyclization tandem reaction of cyclohexane-1,3-diones with β-CF_3_-substituted nitrostyrenes **31** (Figure 12) [71]. It is noteworthy that the cyclization of transient nitrodiketones **32** occurred under very mild conditions (−10 °C) compared to related substrates **27** considered above (cf. data in Figure 10 and Figure 12). This suggests that the squaramide catalyst may be involved in the cyclization stage through the formation of hydrogen bonding with the nitronate moiety.

Linear 1,3-diones react with nitroalkenes **34** in aqueous medium to give 1,5-dihydro-2*H*-pyrrol-2-ones **39** instead of the expected Michael addition products, as demonstrated by Yu et al. (Figure 13) [72]. Unlike the aforementioned transformations, no base was required for this reaction. Note that the use of water as solvent was crucial for the reaction, suggesting that a hydrophobic effect and/or strong hydrogen bonding interactions may be involved in promoting the Michael addition and subsequent cyclization stages. The process was well applicable to a wide variety of nitrostyrenes **34** bearing electron-rich and electron-poor substituents. Aliphatic nitroalkenes were also tolerated, yet the efficiency of the process was much lower. However, nitroalkenes with an *ortho*-substituted aryl group did not give the desired products under these conditions, most likely because of a stronger steric hindrance.

Studies were performed to identify the mechanism of this heterocyclization [72]. NMR monitoring detected the formation of Michael products (nitrodiones **35**) as intermediates. The latter cyclized to furan-2(3*H*)-one oximes **36** (also detected by NMR) through an interrupted Nef/Meyer process. Finally, oximes **36** underwent Beckman-type fragmentation to give nitriles **37**, which recyclized to the final 1,5-dihydro-2*H*-pyrrol-2-ones **39** via amides **38** (Figure 13).

An example of an interrupted Nef/Meyer reaction of cyclic nitronic esters **40** was reported by Ioffe et al. (Figure 14) [73]. On treatment with TfOH, 6-membered cyclic nitronates **40** underwent recyclization to furanone oximes **41** [73]. The suggested mechanism of this process involves the protonation of the nitronate moiety followed by ring opening in cation **I-9** to give protonated nitrile oxide intermediate **I-10** and intramolecular cyclization with the hydroxyl group leading to the product. Note that the iminium cation **I-9** was directly observed by NMR at −40 °C.

Few precedents of the participation of *S*-nucleophiles in the interrupted Meyer reaction were documented in the literature. An early example was reported by Kiprianov and Verbovskaya, who observed that the heating of 2-aminothiophenol with nitroacetic ester [74] or α-nitro ketones [75] afforded 2*H*-benzo[b][1,4]thiazin-2-one oximes **42** and **43**, respectively (Figure 15). This condensation involves the formation of amide or imine intermediates **I-11** and **I-12**, which undergo a spontaneous cyclization via an interrupted Nef/Meyer reaction mechanism. The desired heterocyclic products were formed in moderate yields. In the case of α-nitroacetophenone (R = Ph), 2-phenylbenzothiazole **44** and nitromethane were detected as side products (Figure 15b).

Recently, Batra et al. developed a one-pot protocol for the preparation of thiohydroximic acids **44** from aromatic ketones on treatment with the NaNO_2_/I_2_ system in DMSO (Figure 16) [76]. The suggested mechanism of this multi-stage transformation involved an interrupted Nef/Meyer reaction of α-nitroketones generated in situ. Dimethyl sulfide, which was formed by the deoxygenation of DMSO with HI, served as a nucleophile in the reaction with nitronic acid **I-13**. The subsequent elimination of methanol led to the formation of the final thiohydroximic acid **44**. The method tolerates electron-rich and electron-poor aryl groups, as well as various heteroaromatic rings. However, aliphatic ketones did not give the desired products in this process.

## 4. Interruption of the Nef and Meyer Reactions by *N*-Nucleophiles

In a fashion similar to *O*- and *S*-nucleophiles, some *N*-nucleophiles can interrupt the usual pathway of Nef and Meyer reactions. Examples of these reactions in both inter- and intramolecular modes are known.

In 1964, Bachman and Goldmacher observed the formation of *N*-phenylacetamidoxime via the heating of aniline with nitroethane in polyphosphoric acid (PPA) [77]. This useful reaction is being developed further by Aksenov’s group who reported a general synthesis of *N*-hydroxyimidoamides **46** by condensation of simple nitroalkanes (nitromethane and nitroethane) with anilines **45** in PPA (Figure 17, a) [78]. The products formed in high yields as a mixture of tautomers. Since PPA is both a protic and an electrophilic medium, the reacting cationic iminium species **I-14** can be generated either by protonation or phosphorylation of nitronic acids. Additionally, experimental evidence in support of the intermediacy of hydroxynitrilium cations **I-15** in this reaction was obtained in a later work by the same authors [79].

Using 1,2-phenylenediamines **47** and 2-aminophenols **48** under the same conditions, a cascade heterocyclization process was accomplished (Figure 18) [78]. As a result, a wide range of benzimidazoles **49** and benzoxazoles **50** were obtained with high efficiency.

An intramolecular version of the *N*-interrupted Nef/Meyer reaction was reported by Katritzky et al. (Figure 19) [80]. Here, heating of nitromethyl-substituted benzoxazinones **51** with HCl/dioxane results in their rearrangement to 1,3-benzoxazepines **52** in moderate yields. The suggested mechanism involves the oxazine ring opening followed by an intramolecular cyclization, in which the carbamate nitrogen atom attacks the *N*,*N*-bis(oxy)iminium cation moiety in intermediate **I-16** to form a 7-membered ring. Note that nitroalkene **53** was unreactive under these conditions, suggesting that it is not an intermediate in the process.

A notable example of an intramolecular interrupted Nef reaction was reported by Nishiwaki et al. (Figure 20) [81]. Here, nitro compounds **54** bearing a hydrazone moiety cyclized to give a product in which the structure of nitroso-substituted pyridazines **55** was assigned. Remarkably, the cyclization took place under mild conditions without the need for an external acid. DFT calculations showed the considerably higher electrophilicity of the nitronic acid carbon (C-2) compared to the cyano group (C-1) in intermediate **I-17**. However, the substrate containing an unsubstituted amino group (R = H) cyclized at the C-1 atom leading to a 7-membered cycle.

A remarkable example of a Nef/Meyer reaction interrupted by pyridine was observed by Royer et al. [82] in their studies on the chlorination of electron-rich arenes (such as 2-methoxynaphthalene) with the pyridinium chloride/nitroethane system (Figure 21). It was noted that pyridinium chloride reacted with nitroethane in the presence of free pyridine to give *N*-acethydroxamylation product **56**. The nucleophilic addition of pyridine to the *N*,*N*-bis(oxy)iminium cation **I-18** derived from *aci*-nitroethane was assumed as a reasonable pathway for this reaction. The resulting pyridinium salt **56** is more likely to be a side product in the aforementioned chlorination of arenes rather than an intermediate. Unfortunately, this reaction was not developed any further.

## 5. Interruption of the Nef and Meyer Reactions by *C*-Nucleophiles

A few examples of Nef and Meyer reactions interrupted by some *C*-nucleophiles have been reported in the literature. However, only the *C*-nucleophiles which are compatible with strong protic acids can be applied. Examples of these processes are limited to electron-rich arenes and some heteroarenes as *C*-nucleophiles.

Jacquesy et al. [83] and Shudo et al. [84] showed that the reaction of nitroalkane salts **1** with arenes in an acid medium leads to oximes of substituted acetophenones **57** (Figure 22a). In the case of toluene, phenol, and anisol, mixtures of *ortho*- and *para*-isomers were obtained. Free nitroalkanes [85] and α-carbonylnitromethanes [84,86] can undergo the same reaction in triflic acid (Figure 22b). Under these conditions, the corresponding oximes were prepared from fluorobenzene and naphthalene. It was shown that the process led to a single oxime isomer, in which the entering nucleophile and the lone pair on nitrogen that is formed are arranged *trans* with respect to each other [86]. Moreover, dimers of nitrile oxides (furoxans) were isolated as side products in some cases. Based on this data, the mechanism involving the participation of hydroxynitrilium ions **I-3** and subsequent Friedel−Crafts reaction was suggested [86].

Recently, Aksenov et al. showed that nitroalkanes reacted with electron-rich arenes in polyphosphoric acid medium to form other products. For example, amides of benzoic acids **58** were obtained in the case of nitromethane (Figure 23a) [87], while *N*-arylacetamides **59** formed in the case of nitroethane (Figure 23b) [88]. The suggested mechanism of these reactions includes the electrophilic addition of phosphorylated *N*,*N*-bis(hydroxy)iminium cations **I-19** or **I-20** to the arene, followed by dehydration of oximes to nitriles (in the case of nitromethane, Figure 23a) or by a Beckmann rearrangement (in the case of nitroethane, Figure 23b). As in the examples considered previously (Figure 22), the hydroxynitrilium ions of type **I-3** can also include electrophilic species reacting with arenes.

An intramolecular version of the Meyer reaction interrupted by a *C*-arylation version of this reaction was observed by Yao et al. in the treatment of a salt of sterically hindered nitro compound **60** with sulfuric acid (Figure 24) [45]. Here, benzocyclopentanone oxime **62** was isolated along with the expected hydroxamic acid **61** formed via a normal Meyer reaction. Both reactions may occur via intermediate cation **I-21**, which is corroborated by the fact that the corresponding nitrile oxide was isolated in some experiments starting from nitro compound **60** (cf. Figure 3).

Interestingly, related aryl-substituted nitro carboxylates **63** exhibit an unusual behavior under strong acid conditions as shown by Ohwada et al. [89]. In fact, 3-aryl-2-nitropropionates **63** are converted to 4*H*-1,2-benzoxazines **64** on treatment with trifluoromethanesulfonic acid (Figure 25a). In the case of substrate **63a** with a *para*-methoxyphenyl group, cyclization occurred at the *ipso* position to give a spiro-annulated isoxazoline **65** (Figure 25b). A distinctive feature of the reactions in the examples considered above is that a carbon-oxygen bond with an aryl moiety is formed in the reaction. The exact mechanism of this transformation has not been elucidated; however, as one of the hypotheses, it was assumed that an attack of the nitrosocarbenium ion **I-22** or **I-23** on the aryl fragment took place. The reactivity of nitrosocarbenium ions as formal *O*-electrophiles was also documented by other researchers in reactions of nitronates mediated by strong Lewis acids [90].

## 6. Conclusions

The literature survey revealed a number of processes that can be interpreted as interrupted Nef or Meyer reactions of nitro compounds and nitronates. In these reactions, transient cationic species resulting from the protonation of nitronic acids (aci-nitro) are intercepted with an external nucleophile leading to α-substituted nitroso compounds, hydroxamic acid derivatives, oximes, or products of their further transformations. Halide anions, *O*-nucleophiles (alcohols, enols, carboxylic acids), *N*-nucleophiles (amines, carbamates, hydrazones, pyridine), *S*-nucleophiles (mercaptans), and some *C*-nucleophiles have been shown to redirect the normal pathway of Nef and Meyer reactions in a chemoselective manner. Generally, the nucleophile should be compatible with strongly acidic conditions under which both reactions occur. However, cyclizations in which the nitro moiety is activated by intramolecular hydrogen bonding occur without the need for an external acid. Unlike the normal Nef reaction, its interrupted versions may not require a prior conversion of a nitro compound into its salt.

At the same time, the synthetic potential of interrupted Nef or Meyer reactions has not yet been realized to the full extent. First, the range of nucleophiles involved in these processes is still quite narrow. In particular, the use of *C*-nucleophiles is limited to electron-rich arenes and some heteroarenes. Additionally, the use of *S*-nucleophiles and *N*-nucleophiles appears to be underdeveloped. Secondly, a survey of efficient Brønsted acid and hydrogen-bonding catalysts for interrupted Nef or Meyer reactions is advantageous for the further development of this methodology. The implementation of catalytic methods is expected to broaden the scope of nucleophiles for these processes. Third, examples of the use of cyclic nitronic esters (cyclic nitronates) in interrupted Nef or Meyer reactions are very limited. The use of these readily available substrates may provide straightforward access to polyfunctionalized products possessing a few contiguous stereogenic centers. We feel that progress in these directions can be anticipated in the near future.

Finally, it should be noted that some aspects of the mechanism of the Nef and Meyer reactions are still under debate. In particular, this applies to the participation of putative nitrosocarbenium and *N*-hydroxynitrilium cationic intermediates. An investigation of the interrupted reactions is required to gain further insights into these fundamental problems.

## Data Availability

Not applicable.

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
