# Peer review of "Interrupted Nef and Meyer Reactions: A Growing Point for Diversity-Oriented Synthesis Based on Nitro Compounds"

_molecules, 2023, doi:10.3390/molecules28020686_

Round 1
Reviewer 1 Report
The manuscript presented a concise but thorough summary on the use of interrupted Nef and Meyers reactions in organic synthesis. Compared to the classical Nef and Meyer reactions resulting in carbonyl or carboxylic acid compounds, these transformations allow the approach to more complicated and diversified structures. As a result of these reactions, transient cationic species can interact with various types of nucleophiles instead of water, intervening in the production of carbonyl and carboxylic acids through common Nef/Meyer reactions. Remarkably, the formation of either a-substituted nitroso compounds or oximes is progressively favored when O-, N-, S-, C-nucleophiles, and halide anions were used.
The reactivity using halide ions such as Cl‑, Br-, and I- decreased with an increase in the steric hindrance of the nitronate. Due to the bulkiness of the nitronates, stable nitrile oxides comprise the major product as opposed to hydroxyl halides. Regarding O-, and S-nucleophiles, the intramolecular process occurred at an increased rate in comparison to the parent reaction with water as the nucleophile. Significant contributions to the enol moiety from aliphatic nitro compounds are possible with the presence of strong neighboring groups. Regarding the C-nucleophile, the compatibility of electron-rich arenes and some heteroarenes with strong protic acids are strong. Cyclization mostly occurred when the nitro moiety is activated by intramolecular hydrogen bonding and occurs without the need for an external acid. Although the use of these nucleophiles appears to be in an early stage, further insights into polyfunctionalized products have been elucidated in these studies.
In my opinion, this review is a good resource for those working in organic synthesis, especially total synthesis of complex molecules. Hence, it can be published with consideration of the following comments.
(1) A typo “hydrolytric” (line 161, page 6) should be corrected as “hydrolytic”
(2) Scheme 9 and Scheme 14 were not mentioned in the accompanying texts (lines196-202, page 8 and lines 259-266, page 10-11). It is better to mention them in the text.
(3) When discussing about examples relating to hydroxynitrilium ions of type I-3, the author addressed its appearance in Scheme 21 while the right place should be Scheme 22. Please check (line 391, page 16)
(4) An incorrect assignment of the scheme with accompanying text was detected (line 413, page 17).
(5) The structure of side product furoxans (lines 376-377, page 15) should be added to the associated Scheme.
(6) Two examples on the use of interrupted Nef cascade to synthesize natural products bearing bicycloketal frameworks were given in the manuscript. If (possible) more examples exemplifying the exploration of interrupted Nef and Meyer reactions in the synthesis of valuable compounds can be given, that would make the review more fantastic.
(7) In this regard, the authors should consider the use of Nef-like reactions in the synthesis of heterocycles leading to certain tetrapyrroles including an interrupted Nef reaction (Organics 2022, 3, 262–274). Some mechanistic studies with use of isotopic labeling have been done also that seem to bear on the interruption of the Nef reaction. This work on offshoots of the Nef reaction dates to a paper about 50 years ago by McMurry (J. Org. Chem. 1973, 38, 4367–4373) – shortly predating his McMurry method of alkene synthesis.
(8) In scheme 2c, the temperature is -78 ºC, but page 4, line 95, -40 ºC.
(9) In scheme 4, what is the base for the conversion from 8 to 9?
(10) Scheme 1, please be sure that the arrow is originated from oxygen atom.
(11) Page 2, line 51, ‘X’ was used to represent halides, but the other pages ‘Hal’ was used to represent halide anions and it was non-universal fashion.
(12) Scheme 3, the text under compound 7, ‘Hal = I’ does not explain the formation of compound at the bottom right corner. Compound 7 was formed when the halide was chlorine, not iodine.
(13) For the interruption of the Nef and Meyer reactions by halide anions (topic 1), it would be more interesting if the data is sufficient to draw a general trend of halides’ reactivity towards the interrupted Nef/Meyer reactions.
(14) Scheme 5, wedged bond should be drawn in compound 11 and 12a’/12a”
(15) Page 5, line 140, an O-enolate is also documented and should be included in the first sentence.
(16) Page 7, paragraph 1, according to the sentence “Interestingly, an acid medium is not needed in these reactions, probably due to an intramolecular activation of the nitronate moiety through hydrogen bonding with the carboxylic group”
(17) Drawing of hydrogen bonding interaction in scheme 8 might aid readers’ understanding how the H-bond is formed.
(18) Is charge interaction between nitronate moiety and carboxylate ion more likely to occur than the H-bond?
(19) Please justify that base is not required as well.
(20) Page 7, line 187, the author stated “elimination of a water molecule” but the chemical formula of compound 21 and 22 is identical. Loss of water is unbelievable.
(21) Page 8, paragraph under scheme 8, please clarify that acid is not needed.
(22) Scheme 8a, elimination of N2O does not make sense. I-8 has only one nitrogen.
(23) Page 10, paragraph 1, please add some sentence about what could drive the reaction forward if a base was not demanded to abstract the proton to generate the enolate.
(24) For the interruption of the Nef and Meyer reactions by N-nucleophiles (topic 4), the general Nef and Meyer reactions in scheme 1 cannot explain the formation of transient cationic species because bases are not used here. A general scheme showing how the cationic species are formed under PPA treatment might be more informative.
(25) Scheme 7, 10, 11, 15, 17, 19, 20, 21, and 23, loss of small molecules should be added.
(26) Scheme 18 and 23, what does “tº” stand for?
(27) Last sentence of the paper. The use of “will” is prophetic, and “will definitely” makes the prophesy more emphatic. Better scientific writing would be: “Investigation of interrupted reactions is required to gain further insights….”
References:
(1) For the reference format, please be consistent in accordance with the requirement of the journal. In the current version of the manuscript, the name of cited papers is presented in both ways (uncapitalizing and capitalizing each word). Please fix.
- Correct papers’ title that has a superscript, subscript, or italic letter (Ref. 26, 37, 49, 51, 52, 58, 61, 62, 64, 66, and 67)
- Ref. 43, remove DOI.
Some comments on writing:
1. Page 2, line 52, “nitro group was” should be “nitro group is”
2. Page 2, line 60, “analyses” should be “analyzes”
3. Page 3, line 94, “reactions of sodium salts” should be “reactions of sodium salts”
4. Page 3, line 102, “Cl-, Br-, I- dropped” should be “of Cl-, Br- and I- decreased”
5. Page 4, line 106, “green color” should be “green colors”
6. Page 4, line 108, “can operate” should be “may operate”
7. Page 5, line 126, “treatment of the later” should be “treatment of the latter”
8. Page 5, line 127, “in ring opening” should be “in the ring opening”
9. Page 5, line 141, “all these transformations” should be “all of these transformations”
10. Page 7, line 170, “probably due to” should be “likely due to”
11. Page 7, line 183, “on treatment” should be “after treatment”
12. Page 9, line 213, “as resulting from” should be “as the result of”
13. Page 10, line 243, “unlike in the” should be “unlike the”
14. Page 10, line 259, “of interrupted” should be “of an interrupted”
15. Page 11, line 275, “products formed” should be “products were formed”
16. Page 12, line 300, “on heating” should be “via heating”
17. Page 12, line 301, “recently developed” should be “is being developed”
18. Page 14, line 335, “to which” should be “in which”
19. Page 14, line 339, “containing unsubstituted” should be “containing an unsubstituted”
20. Page 14, line 350, “pathway of” should be “pathway for”
21. Page 16, line 390, “like in” should “be as”
22. Page 17, line 433, “acid conditions” should be “acidic conditions”
23. Page 17, line 434, “an intramolecular” should be “intramolecular”
24. Page 17, line 446, “a straightforward” should be “straightforward”
25. Use commas to separate three or more words in a series. Before ‘and’ should add an Oxford comma.
26. Page 8, line 212, a full stop should be added after number 29.
In summary, this article is suitable for publication on the Molecules with revisions.
Author Response
Response to Reviewer 1 Comments
Thank you for a positive evaluation of my manuscript and your very helpful comments. Answers to your comments are given below:
Reviewer’s comment (#1): “A typo “hydrolytric” (line 161, page 6) should be corrected as “hydrolytic””
Answer: Thank you. This mistake was corrected.
Reviewer’s comment (#2): “Scheme 9 and Scheme 14 were not mentioned in the accompanying texts (lines196-202, page 8 and lines 259-266, page 10-11). It is better to mention them in the text.”
Answer: Thank you. References to these Schemes were added in the text.
Reviewer’s comment (#3): “When discussing about examples relating to hydroxynitrilium ions of type I-3, the author addressed its appearance in Scheme 21 while the right place should be Scheme 22. Please check (line 391, page 16)”
Answer: Thank you. This mistake was corrected.
Reviewer’s comment (#4): “An incorrect assignment of the scheme with accompanying text was detected (line 413, page 17).”
Answer: Thank you. This mistake was corrected.
Reviewer’s comment (#5): “The structure of side product furoxans (lines 376-377, page 15) should be added to the associated Scheme.”
Answer: Thank you. The structure of side product (furoxan) was added in Scheme 22.
Reviewer’s comment (#6): “Two examples on the use of interrupted Nef cascade to synthesize natural products bearing bicycloketal frameworks were given in the manuscript. If (possible) more examples exemplifying the exploration of interrupted Nef and Meyer reactions in the synthesis of valuable compounds can be given, that would make the review more fantastic.”
Answer: Thank you for this suggestion. Unfortunately, applications of intrerrupted Nef reactions in total synthesis are scarce. Probably, there will be more in future, and I hope this review will stimulate research in this direction. Anyway, one more example was found in the literature (total synthesis of rac-pederamide, J. Am. Chem. Soc. 1979, 101, 5364-5370), which was overlooked in the previous literature survey. A discussion of this synthesis and the corresponding Scheme (Scheme 6) were added on page 6. Also, structures of the synthesized natural products were added in Scheme 7 (former Scheme 6 in the original manuscript) and mentioned in the revised text.
Reviewer’s comment (#7): “In this regard, the authors should consider the use of Nef-like reactions in the synthesis of heterocycles leading to certain tetrapyrroles including an interrupted Nef reaction (Organics 2022, 3, 262–274). Some mechanistic studies with use of isotopic labeling have been done also that seem to bear on the interruption of the Nef reaction. This work on offshoots of the Nef reaction dates to a paper about 50 years ago by McMurry (J. Org. Chem. 1973, 38, 4367–4373) – shortly predating his McMurry method of alkene synthesis.”
Answer: These papers correspond to a reductive version of the Nef reaction (the McMurry-Melton reaction). However, as stated in the introduction, this mini-review is focused on the interrupted reactions, which take place under standard Nef conditions without external reducing or oxidizing reagents. Anyway, these two papers were cited in the revised introduction section together with other key developments in the field of the Nef reaction and related processes.
Reviewer’s comment (#8): “In scheme 2c, the temperature is -78 ºC, but page 4, line 95, -40 ºC.”
Answer: Thank you. This mistake was corrected in the text. The correct temperature is -78 ºC.
Reviewer’s comment (#9): “In scheme 4, what is the base for the conversion from 8 to 9?”
Answer: The authors did not use any base in this reaction. In fact, many interrupted Nef reactions do not require the pre-generation of the nitronate salt (unlike normal Nef reaction, see also answer to comment #24). A comment on this was added in the conclusion section.
Reviewer’s comment (#10): “Scheme 1, please be sure that the arrow is originated from oxygen atom.”
Answer: Thank you. This was corrected.
Reviewer’s comment (#11): “Page 2, line 51, ‘X’ was used to represent halides, but the other pages ‘Hal’ was used to represent halide anions and it was non-universal fashion.”
Answer: Thank you. This was a mistake, “C-X bond” in page 2 corresponds carbon-heteroatom bond. The text was corrected accordingly.
Reviewer’s comment (#12): “Scheme 3, the text under compound 7, ‘Hal = I’ does not explain the formation of compound at the bottom right corner. Compound 7 was formed when the halide was chlorine, not iodine.”
Answer: Thank you. The scheme was corrected and some more representative examples were added to showcase the dependence of the chemoselectivity on the sterical hindrance of the substrate and the nature of nucleophile.
Reviewer’s comment (#13): “For the interruption of the Nef and Meyer reactions by halide anions (topic 1), it would be more interesting if the data is sufficient to draw a general trend of halides’ reactivity towards the interrupted Nef/Meyer reactions.”
Answer: Thank you for this suggestion. Unlike hydroxyimidoyl chlorides, the yields of the corresponding bromides and especially iodides substantially decreased with an increase in the steric hindrance of the nitronate. A comment on this was added in the text.
Reviewer’s comment (#14): “Scheme 5, wedged bond should be drawn in compound 11 and 12a’/12a””
Answer: Thank you. This was corrected.
Reviewer’s comment (#15): “Page 5, line 140, an O-enolate is also documented and should be included in the first sentence.”
Answer: Thank you. “O-enolate” was included in the first sentence.
Reviewer’s comment (#16): “Page 7, paragraph 1, according to the sentence “Interestingly, an acid medium is not needed in these reactions, probably due to an intramolecular activation of the nitronate moiety through hydrogen bonding with the carboxylic group””
Answer: There seem to be no changes required according to this comment.
Reviewer’s comments (##17-19): “Drawing of hydrogen bonding interaction in scheme 8 might aid readers’ understanding how the H-bond is formed. Is charge interaction between nitronate moiety and carboxylate ion more likely to occur than the H-bond? Please justify that base is not required as well.”
Answer: It seems to be an ambiguous question. Nitronic esters (and so the nitronic acids) are poor C-electrophiles. Activation by protonation of the N-oxide oxygen (or addition of a Lewis acid) is needed to promote further nucleophilic addition. It can be assumed that hydrogen bond donors may also activate nitronic acids towards addition of nucleophiles in a similar manner. Intramolecular hydrogen bonds are known to occur in nitroalcohols, yet any experimental evidence for nitrocarboxylic acids could not be found. However, in the case of intramolecular cyclization of nitrocarboxylic acids to I-7 (Scheme 9), the plausible formation of a hydrogen bond, charge interaction and nucleophilic addition are not separated in time and take place simultaneously. In order to avoid any speculation, the zwitterionic intermediate was removed from the top part of Scheme 9 (former Scheme 8) and mention of hydrogen bonding in the case of nitrocarboxylic acids was avoided.
Reviewer’s comment (#20): “Page 7, line 187, the author stated “elimination of a water molecule” but the chemical formula of compound 21 and 22 is identical. Loss of water is unbelievable.”
Answer: Thank you for pointing on this issue. The loss of water corresponds to the cyclization of nitroacid 20 to the compound 21. I’m sorry for this misleading. The text and the Scheme were corrected accordingly.
Reviewer’s comment (#21): “Page 8, paragraph under scheme 8, please clarify that acid is not needed.”
Answer: Unlike the cyclization of nitrocarboxylic acids, the cyclization of compound 25 (which is a nitroamide) proceeds in the presence of acid (AcOH). The sentence was revised to highlight this fact. Also, a comment on the participation of nitrile oxides in this process was added together with a reference to another work describing a similar process.
Reviewer’s comment (#22): “Scheme 8a, elimination of N2O does not make sense. I-8 has only one nitrogen.”
Answer: Thank you for pointing on this issue. “N2O” was changed to “HNO” in this reaction equation.
Reviewer’s comment (#23): “Page 10, paragraph 1, please add some sentence about what could drive the reaction forward if a base was not demanded to abstract the proton to generate the enolate.”
Answer: The use of water as solvent is crucial for this reaction suggesting that a hydrophobic effect and/or strong hydrogen bonding interactions may be involved in promoting the Michael addition and subsequent cyclization stages. A comment on this was added in the text.
Reviewer’s comment (#24): “For the interruption of the Nef and Meyer reactions by N-nucleophiles (topic 4), the general Nef and Meyer reactions in scheme 1 cannot explain the formation of transient cationic species because bases are not used here. A general scheme showing how the cationic species are formed under PPA treatment might be more informative.”
Answer: Thank you for this comment. Remarkably, many of the reported interrupted Nef reactions do not require the initial conversion of nitro compounds into nitronate salts and proceed through a direct activation of the nitroalkane fragment with a protic acid (see Schemes 4, 9, 11c, 13, 15-20, 22b, 23 and 25). This situation is different from the normal Nef reaction, albeit few examples of direct acidic hydrolysis of nitroalkanes into aldehydes/ketones (without a prior deprotonation) are known in the literature (for example, see Can. J. Chem. 1971, 49, 3493; J. Am. Chem. Soc., 1962, 84, 688). It seems that under conditions used in some interrupted Nef reactions, the electrophilic species are efficiently generated through tautomerization of nitro compounds and subsequent protonation of aci-forms. This interesting fact was specially highlighted in the revised conclusion section. Also, Scheme 1 was modified to include tautomerization of a nitroalkane to aci-form and its protonation to give iminium cation I-1.The generation of electrophilic species from nitroalkanes in polyphosporic acid (PPA) is not obvious. The mechanism of PPA-promoted activation of nitroalkanes was not sufficiently studied. PPA is known to be an amphoteric medium (Energy Fuels, 2008, 22, 4, 2637), and it also can serve as an electrophilic phosphorylating reagent. In principle, PPA may facilitate tautomerization of nitroalkanes and convert them into O-phosphorylated nitronates, phosphorylated N,N-bis(oxy)iminium species as well as nitrile oxides. It is not clear which of these species are intermediates in the interrupted Nef reaction in PPA medium. In order to avoid speculations, I decided not to give any general scheme for the generation of electrophilic species from nitroalkanes in PPA.
Reviewer’s comment (#25): “Scheme 7, 10, 11, 15, 17, 19, 20, 21, and 23, loss of small molecules should be added.”
Answer: Thank you for pointing on this issue. Elimination of small molecules was added in these Schemes.
Reviewer’s comment (#26): “Scheme 18 and 23, what does “tº” stand for?”
Answer: Thank you for pointing on this issue. Temperature was specified in these Schemes.
Reviewer’s comment (#27): “Last sentence of the paper. The use of “will” is prophetic, and “will definitely” makes the prophesy more emphatic. Better scientific writing would be: “Investigation of interrupted reactions is required to gain further insights….””
Answer: Thank you. The sentence was corrected accordingly.
Reviewer’s comment on references (#1): “For the reference format, please be consistent in accordance with the requirement of the journal. In the current version of the manuscript, the name of cited papers is presented in both ways (uncapitalizing and capitalizing each word). Please fix. - Correct papers’ title that has a superscript, subscript, or italic letter (Ref. 26, 37, 49, 51, 52, 58, 61, 62, 64, 66, and 67) - Ref. 43, remove DOI.”
Answer: Thank you. The reference format was corrected as per journal requirements. Names of citing papers were given without capitalizing each word. Text formatting (superscript, subscript, italic letters) was checked in titles of articles. Page and issue numbers were added for the newly published paper in Ref 43 (ref. 49 in the revised manuscript).
Reviewer’s comments on writing (#1,2,4-26).
Answer: Thank you for pointing on these writing issues. These mistakes were corrected as per your suggestions.
Reviewer’s comment on writing (3): “. Page 3, line 94, “reactions of sodium salts” should be “reactions of sodium salts””
Answer: There seem to be no changes required according to this comment.

Reviewer 2 Report
The author summarized the development of interrupted Nef and Meyer reactions by various nucleophiles, and the prospects of this methodology in the diversity-oriented organic synthesis were analyzed. The manuscript is well-organized and well-written. According to the types of the nucleophiles, the main text is divided into the following four parts: 1. Interruption of the Nef and Meyer reactions by halide anions; 2. Interruption of the Nef and Meyer reactions by O- and S-nucleophiles; 3. Interruption of the Nef and Meyer reactions by N-nucleophiles; 4. Interruption of the Nef and Meyer reactions by С-nucleophiles. But, the author have not mentioned these four types of nucleophiles in introduction section. It is better to briefly introduce these four types of nucleophiles in the introduction section. In addition, the author is advised to introduce and cite the references in chronological order as much as possible in the same section. Some mistakes need to be corrected before the acceptance. This reviewer has found the following mistakes:
1. Page 5, line 126, “later” should be “latter”. The “latter” is a noun, whereas the “later” is an adjective or adverb.
2. Page 12, line 301, there is a “was” missing between the words “reaction” and “recently”.
3. Page 14, line 339, there is an “an” missing between the words “containing” and “unsubstituted”. The absence of indefinite or definite articles seems to be common in this text. This problem should be checked throughout the manuscript.
4. Page 17, line 430, “mercaptans” is misspelled;
Please correct all these mistakes and some of the typo mistakes in the text. The author is also encouraged to check the whole manuscript and find more problems.
Author Response
Thank you for a positive evaluation of my manuscript and your very helpful comments. Answers to your comments are given below:
Reviewer’s comment: “But, the author have not mentioned these four types of nucleophiles in introduction section. It is better to briefly introduce these four types of nucleophiles in the introduction section.”
Answer: Thank you for pointing on this issue. The first and the last paragraphs on page 2 (introduction section) were corrected as per your suggestion. The sub-division of the review material according to these four types of nucleophiles was mentioned in the last paragraph of the revised introduction section.
Reviewer’s comment: “In addition, the author is advised to introduce and cite the references in chronological order as much as possible in the same section.”
Answer: Thank you for this suggestion. The introduction section was reorganized and re-written as per your suggestion. Citations were introduced in a chronological order as much as possible in this section. Also, several new citations of some key works dealing with the development/application of the Nef reaction were added.
Reviewer’s comment: “Page 5, line 126, “later” should be “latter”. The “latter” is a noun, whereas the “later” is an adjective or adverb.”
Answer: Thank you. This mistake was corrected.
Reviewer’s comment: “Page 12, line 301, there is a “was” missing between the words “reaction” and “recently””
Answer: Thank you. This mistake was corrected.
Reviewer’s comment: “Page 14, line 339, there is an “an” missing between the words “containing” and “unsubstituted”. The absence of indefinite or definite articles seems to be common in this text. This problem should be checked throughout the manuscript.”
Answer: Thank you. This mistake was corrected. The use of articles was checked and corrected across the manuscript. Also, some grammar mistakes found by Reviewer 1 were also corrected.
Reviewer’s comment: “Page 17, line 430, “mercaptans” is misspelled”.
Answer: Thank you. This mistake was corrected.
